A successful method to restore seagrass habitats in coastal areas affected by consecutive natural events

Ruiz-Diaz Claudia Patricia claudiaruiz@sampr.org
Toledo-Hernández Carlos
Sánchez-González Juan Luis
Mercado-Molina Alex E.
Sociedad Ambiente Marino , San Juan , Puerto Rico
Mahmood Haider
Electronic publication date: 2024 Jan 2
Publication date: 2024
Volume: 12
Electronic Location ID: e16700
Received 2023 Sep 18; Accepted 2023 Nov 29
Copyright: © 2024 Ruiz-Diaz et al.
Copyright year: 2024
Copyright holder: Ruiz-Diaz et al.
License: This is an open access article distributed under the terms of the Creative Commons Attribution License, which permits unrestricted use, distribution, reproduction and adaptation in any medium and for any purpose provided that it is properly attributed. For attribution, the original author(s), title, publication source (PeerJ) and either DOI or URL of the article must be cited.
License URL: https://creativecommons.org/licenses/by/4.0/

Keywords: Thalassia testudinum, Syringodium filiforme, Plug propagation units (PUS), Hurricanes, Halophila stipulacea, Caribbean, Carbon storage, Restoration

Funding: NOAA award NA20NMF4630303 This study was supported by NOAA award NA20NMF4630303. The funders had no role in study design, data collection and analysis, decision to publish, or preparation of the manuscript.

==============================
Background

Seagrass meadows, known for providing essential ecosystem services like supporting fishing, coastline protection from erosion, and acting as carbon sinks to mitigate climate change effects, are facing severe degradation. The current deteriorating state can be attributed to the combination of anthropogenic activities, biological factors (i.e., invasive species), and natural forces (i.e., hurricanes). Indeed, the global seagrass cover is diminishing at an alarming mean rate of 7% annually, jeopardizing the health of these vital ecosystems. However, in the Island Municipality of Culebra, Puerto Rico, losses are occurring at a faster pace. For instance, hurricanes have caused over 10% of cover seagrass losses, and the natural recovery of seagrasses across Culebra’s coast has been slow due to the low growth rates of native seagrasses (Thalassia testudinum and Syringodium filiforme) and the invasion of the invasive species Halophila stipulacea. Restoration programs are, thus, necessary to revitalize the native seagrass communities and associated fauna while limiting the spread of the invasive species.

Methods

Here, we present the results of a seagrass meadow restoration project carried out in Punta Melones (PTM), Culebra, Puerto Rico, in response to the impact of Hurricanes Irma and María during 2017. The restoration technique used was planting propagation units (PUs), each with an area of 900 cm2 of native seagrasses Thalassia testudinum and Syringodium filiforme, planted at a depth between 3.5 and 4.5 m. A total of 688 PUs were planted between August 2021 and August 2023, and a sub-sample of 88 PUs was monitored between August 2021 and April 2023.

Results

PUs showed over 95% of the seagrass survived, with Hurricane Fiona causing most of the mortalities potentially due to PUs burial by sediment movement and uplifting by wave energy. The surface area of the planting units increased by approximately 200% (i.e., 2,459 cm2), while seagrass shoot density increased by 168% (i.e., 126 shoots by PU). Additionally, flowering and fruiting were observed in multiple planting units, indicating 1) that the action taken did not adversely affect the PUs units and 2) that the project was successful in revitalizing seagrass populations. The seagrass restoration project achieved remarkable success, primarily attributed to the substantial volume of each PUs. Likely this high volume played a crucial role in facilitating the connection among roots, shoots, and microfauna while providing a higher number of undamaged and active rhizome meristems and short shoots. These factors collectively contributed to the enhanced growth and survivorship of the PUs, ultimately leading to the favorable outcome observed in the seagrass restoration project.

Introduction

Seagrass restoration activities have trailed those of coral reefs restoration owing to numerous factors. First, the cost of restoring seagrass meadows tends to be higher than for coral reefs. To illustrate, while restoring 1 hectare of coral reefs is estimated at approximately US $11,717, the restoration of an equivalent area of seagrass meadows has been projected at an astonishing cost of roughly US $2,879,773 (Bayraktarov et al., 2015). Furthermore, the disparity in funding invested between seagrass meadows and coral reefs is also quite impactful. For every US $ invested in seagrass research, US $244.00 is invested in coral research (Orth et al., 2006; Unsworth et al., 2019). The disproportionate attention given to coral reefs over seagrass meadows is also reflected in the number of peer-reviewed publications (Hind-Ozan & Jones, 2018). In a study by Unsworth et al. (2019) that analyzed research publications from 1992 to 2016, it was found that while seagrass publications have shown a linear increase, publications on coral reefs have grown exponentially. To put it in perspective, for every 16 seagrass-related publications, 100 studies focused on coral reefs have been published. Second, the success of seagrass meadow restoration is lower than that of coral reef restoration projects (Bayraktarov et al., 2015).

For instance, the global success rate of seagrass restoration initiatives stands at approximately 30% (Fonseca, Kenworthy & Thayer, 1998; Orth et al., 2006). Thirdly, the slow growth of seagrasses, particularly climax species such as Thalassia testudinum, poses major challenges for restoration efforts. In the case of T. testudinum, its sluggish growth can be attributed to its strong dependence on active apical meristems from rhizomes and short shoots, which have inherently slow growth rates (Tomlinson, 1974; Gallegos et al., 1993; Andorfer & Dawes, 2002; Furman et al. 2019). Hence, their recovery may take several weeks when meristems are injured or cut. As a result, seagrass restoration projects are costly and time-consuming (Tuya et al., 2017; Rezek et al., 2019; Tan et al., 2020). Fourth, seagrasses are less colorful, spatially less complex, and, from a public view, harbor less biodiversity than coral reefs (Nordlund et al., 2018; Orth et al., 2006), making them less appealing for funding.

Seagrass restoration should be a top priority given their current state of degradation worldwide and the multiple socio-economic and ecological services they render to humankind. For instance, seagrass meadows account for 10% of the oceanic capacity to store carbon and are a major source of oxygen in ocean water, yet they occupy only 0.2% of the world’s oceans (Fourqurean et al., 2012; McLeod et al., 2011). Seagrass meadows accrete sediments, which in turn attenuate wave energy, thereby reducing coastal erosion (Mejia et al., 2016; John et al., 2016; Christianen et al., 2013; Guannel et al., 2016). Furthermore, by trapping sediments, seagrasses improve water quality and thus reduce the abundance of waterborne pathogens (León-Pérez, Hernández & Armstrong, 2019; Paul, 2018). Additionally, thousands of species depend on seagrasses for feeding, nursery grounds, and shelter, as is the case of the multiple organisms along their migratory routes (Herrera et al., 2023; Potouroglou et al., 2017; United Nations Environment Programme, 2020) and the endangered green turtle (Chelonia mydas) and the West-Indian manatee (Trichechus manatus). Likewise, predatory reef fish, belonging to the Carangidae, Serranidae, and Lutjanidae families, utilize seagrass meadows as feeding and nursery grounds at various stages of their life cycle. These fishes are crucial in regulating the trophic dynamics of both reefs and seagrass meadows, in addition to being commercially and nutritionally desirable for human beings (Unsworth et al., 2008; Unsworth & Cullen, 2010).

The main objective of this project was to determine how effective the propagation planting units (PUs) method is in restoring seagrass meadows after significant environmental disturbances have disrupted them. PUs has been used with moderate success in sub-tropical Atlantic seagrass meadows (Fonseca, Kenworthy & Thayer, 1998). However, as far as we know, this method has never been implemented in seagrass meadows dominated by T. testudinum in the Caribbean. Specifically, this study is a pilot project aimed at restoring two acres of seagrass meadow in the Canal de Luis Peña Natural Reserve (CLPNR), Culebra, Puerto Rico. In recent years, seagrass meadows at CLPNR have been impacted by multiple environmental disturbances, including two five-category hurricanes, Irma (Sept. 6, 2020) and María (Sept. 20, 2020). These hurricanes led to a 10% decrease in the seagrass cover due to storm waves and runoff from the intense rain buried (Hernández-Delgado et al., 2020; Toledo-Hernández et al., 2018). Additionally, the rapid expansion of Halophila stipulacea, an invasive seagrass from the Red Sea and Indo Pacific (Ruiz & Ballantine, 2004; Ruiz, Ballantine & Sabater, 2017), has made it the dominant seagrass in some areas formally occupied by native seagrasses before the hurricanes (Hernández-Delgado et al., 2020).

Materials and Methods

Study area: The restoration project was carried out at Punta Melones (PTM), an area within the CLPNR severely damaged by the impacts of the hurricanes. The Puerto Rico Department of Natural and Environmental Resources approved this study (Number DRNA: 2020-IC-020; O-VS-PVS15-SJ-01104-03122019). In general, CLPNR covers nearly 475 hectares. Inland borders of CLPNR are characterized by having low permanent human settlements, no agricultural activities, and a coastal tropical forest nearly in pristine conditions, although human development in recent years is starting to build up (NER-HFA, 2023). Therefore, no major runoffs drain into the coast, and consequently, nearshore waters next to the reserve are clear year-round (Ruiz-Diaz et al., 2016; Toledo-Hernández, Sabat & Zuluaga-Montero, 2007; Hernández-Delgado & Ortiz-Flores, 2022). CLPNR is unique in that it harbors one of the healthier and best well-preserved reefs/hard grounds ecosystems and seagrass meadows dominated by T. testudinum and S. filiforme. Seagrass cover at PTM ranged from 77% to 90% coastline with T. testudinum and S. filiforme and some patches of H. stipulacea (Fig. 1).

Figure 1 Restoration Area.

(A) Map of Culebra illustrating its geographic location with respect to the Caribbean Region. (B) Indicates Puerto Rico and Culebra Island. (C) The red circle indicates the Canal Luis Peña Natural Reserve study site on Culebra Island. Map data: Esri 2023.

Planting propagation units: A total of 688 PUs, each of ~30 cm × 30 cm × 30 cm (length × width × height), dominated primarily by T. testudinum and, to a lesser extent, S. filiforme, were planted at PTM. PUs were harvested following Fonesca, Kenworthy & Thayer (1987) (Fig. 2A), with some modifications. For instance, Fonesca, Kenworthy & Thayer (1987) used a rounded core sampler with a diameter and a length of 20 cm to extract PUs from donor seagrass meadows. Additionally, they employed wire anchors measuring 20 cm in length and bent into U-shaped pins to secure the PUs to the ground. We, on the other hand, extracted the PUs from donor beds using a stainless-steel peat pot with dimensions of approximately 30 cm × 30 cm × 30 cm (length × width × height). To plant the PUs, we excavated pits of sufficient depth to level the PUs’ areal vegetation with the surrounding ground, eliminating the need for anchors to secure them (Figs. 2A–2C). This method allows for the collection of PUs with their associated sediments, roots, shoots, and fauna, providing the PUs with higher chances of survival and growth (Fig. 2A). PUs harvesting was conducted in ten donor seagrass meadows within CLPNR. The distance between the donor sites and the restoration area varied from 200 to 1,000 m, but the collection depth was similar to that of the restoration area (~5 m of water depth). PUs collected at a distance greater than 400 m from the restoring area were brought to a vessel and placed in a container of 102.0 liters filled with water and then taken to the restoring area (less than a 15 min boat ride). Donor seagrass beds have been surveyed since 2019 (Toledo-Hernández & Ruiz-Diaz, 2022), therefore these sites were selected based on (1) their high abundance T. testudinum i.e., at least 85% dominance from T. testudinum. (2) H. stipulacea has not been sighted at the area; 3) no major human-derived stress such as vessel anchoring or environmental perturbation such as land runoff has been observed after the 2017 hurricanes. As a result, the selected sites have a high likelihood of successful recovery after the extraction of PUs. Nonetheless, the scars produced by the collection of PUs were backfilled with biodegradable hessian bags of 30 to 60 kg and filled with sand to accelerate the natural recolonization of seagrass from the scars borders. Because asexual propagation is the primary mode of reproduction of the native seagrasses, collecting material from different donor sites should increase the genetic diversity within the restored site, thereby increasing the probability of persistence during physical, biological, and anthropogenic disturbances. After harvesting, PUs were planted in the sandy area between the reef and the seagrass bed on the same day of collection. The select areas were covered by seagrasses before the hurricane’s impact. Four planting events were carried out, i.e., August 2021, February 2022, March 2022, and May 2022. During each planting event, 22 PUs were tagged with plastic numbered labels attached to a metallic stake and fastened next to the PU for a total of 88 tag PUs. To facilitate the monitoring, PUs from the different planting events were clustered within restoring area in blocks, where block A (BA) included the PUs planted in August 2021 and included six monitoring visits, block B (BB) included PUs planted in February 2022 and were visited four times, block C (BC) included PUs planted in March 2022 and four monitoring visits, and block D (BD) included PUs planted in May 2022 and included four monitoring visits. BD PUs were planted within a monospecific stand of H. stipulacea. All PUs were planted between 1 to 1.5 m apart from each other at a depth of ~5 m.

Figure 2 Plug propagation Units (PUs) in the field.

(A) Show the PU ready to be planted. (B) PU area delimitation. (C) Block of PUs in the initial period of planting. (D) Growth of two PUs where it can be seen that their shoots have joined together.

During each monitoring session, we assessed the survival and growth of each tagged PUs. Survival was defined as PUs with at least one shoot. To determine the percentage of seagrass cover within each PU, we first delimited the area of the PUs, and then estimated the percentage covered by seagrasses within it. Such observations allowed us to assess the degree to which PUs have spread across their surroundings. We also recorded the shoot density specific to each tagged PU by counting the number of shoots within the estimated area. This method helped us to estimate the occurrence of new shoots. Additionally, we measured the length of 10 leaves (in cm) by selecting ten blades at random from within each tagged PU. This permitted us to detect structural changes occurring in PUs over time (Fig. 2).

Analysis: We performed a Pearson’s rank correlation test to determine the relationship between the area of PUs and their corresponding shoot density. Furthermore, we conducted a Kruskal-Wallis test to assess whether Hurricane Fiona had a significant impact on the PUs’ area and shoot density before and after the hurricane. This statistical analysis was carried out using RStudio Team V2021.09.0, and the graphical representations were generated using the ggplot2 package (R Core Team, 2021).

Results

Out of the 88 PUs that were tagged, 38 of them lost their tags by the end of the study in April 2023. As a result, we were unable to monitor them throughout the entire study period. It is likely that these tags were lost due to storm surges caused by Hurricane Fiona, which hit Culebra in September 2022. However, despite this, we found that 95% (657 out of 688) of the PUs survived until the end of the study. Such a high survival rate suggests that most, if not all, of the originally tagged PUs may have survived. Overall, from the PUs tagged, the total area covered at the end of the experiment increased by 1,796.49 cm2 with respect to the initial area. PUs at BA increased in area by 231.72%, 619 days after transplantation. The mean PUs area in BB increased by 344.06% after 428 days. For BC, with 385 days elapsed, the mean covered area increased by 102.69%, while the PUs area from BD, 341 days after transplanted, increased by 119.95% with respect to their initial area, (Fig. 3 and Table 1).

Figure 3 Area of PUs through time in cm2.

(A) Block A, the first PUs planted and tagged. (B) Block B, the second PUs planted and tagged. (C) Block C, the third block planted and tagged. (D) Block D, the fourth block planted and tagged.

Table 1 Monitoring of plug propagation units (PUs).

Block	Date	Area cm2	Shoots #	Shoots (cm2day)−1	Covert %	
BA	Aug.-21 (planted)	1195.14	59		72.34	
BA	Feb-22	1489.22	75	1.33E−06	74.11	
BA	May-22	1793.05	76	2.88E−07	67.24	
BA	Jul-22	2356.83	89	1.08E−05	70.28	
BA	Feb-23	2259.19	77	2.55E−06	63.76	
BA	Apr-23	3964.50	83	8.64E−07	60.42	
BB	Feb.-22 (planted)	1303.13	84		69.37	
BB	Jul-22	2044.76	105	1.36E−05	75.31	
BB	Feb-23	1970.92	153	−3.31E−04	69.98	
BB	Apr-23	5786.75	174	9.19E−07	69.69	
BC	Apr.-22 (planted)	886	47		79.50	
BC	Jul-22	1075.8	55	3.77E−05	66.00	
BC	Feb-23	1535.71	77	3.22E−05	69.10	
BC	Apr-23	1795.83	85	1.61E−05	69.50	
BD	May.22 (planted)	1396.71	110		91.00	
BD	Jul-22	2524.43	106	−6.44E−05	80.29	
BD	Feb-23	1691.29	143	2.04E−04	77.29	
BD	Apr-23	3072.14	163	2.24E−04	60.00	
Note:

Monitoring Plug Propagation Units (PUs) area in cm2, shoots density, shoots day cm2, and cover % for each block over time mean.

Overall, 70.60% of the tagged PUs showed an increment in shoot density compared to the initial density (Fig. 4). From this, 72% of the PUs showed a constant increase in shoot development across the monitoring periods. The shoot density increased by 137.36% with respect to the number of initial shoots. However, the remaining PUs that decreased in their shoot density did it by 42.67%. When analyzing shoot density by blocks, BA increased by 40.17% with a rate of 8.46E-3 shoot*(cm2day)−1. BB increased by 105.53%, i.e., 2.47E-2 shoot*(cm2day)−1. BC increased 81.12%, that is nearly 1.35E-2 shoot*(cm2day)−1. BD increased by 48.27%, i.e., 1.97E-2 shoot*(cm2day)−1 (Figs. 4B and 4C, Table 1).

Figure 4 Total shoot density by PU Thalassia testudinum shoots plus Syringodium filiforme shoots in each PUs through time by blocks.

(A) Showed Block A. (B) Showed Block B. (C) Showed Block C, and (D) Showed Block D.

The Pearson analysis tests revealed a strong positive and significant correlation between the increase in area covered by PUs and the increase in shoot density (Table 1). Therefore, the gross area of the PUs is a reflection of the new shoots through time (Fig. 5). However, the change in PUs area, and consequently shoot density, was not evenly distributed throughout the PUs (Figs. 2B, 3 and 4). The uneven PUs growth caused a reduction in their cover of 64.93% by the end of the monitoring period, i.e., April 2023 (Fig. 6). The reduction in % cover was observed in all blocks, yet BB and BC showed the highest reduction in cover, with 69.82% and 69.50%, respectively (Figs. 6B and 6C). The other blocks, BA and BD, reached 60.42% and 60.00%, respectively (Figs. 6A–6D).

Figure 5 Correlation between the total shoots and the area by each PU by blocks.

The regression analysis shows a significant positive correlation between the total shoots (Thalassia testudinum plus Syringodium filiforme) and the area by each PU by blocks. (A) Showed Block A. (B) Showed Block B. (C) Showed Block C. (D) Showed Block D.

Figure 6 Percentage of seagrass cover within PUs through monitoring date.

(A) Showed Block A. (B) Showed Block B. (C) Showed Block C. (D) Showed Block D.

In general, the height of the PUs canopy increased by 23% at the end of the monitoring, i.e., from an initial measurement of 10.26 cm ± 3.59 to a final measurement of 12.07 cm ± 5.75 (Fig. S1). When the height was analyzed by blocks, three of the four blocks showed a net increase in the canopy. For instance, the canopy at BA increased its height by 2% at the end of the monitoring, and BC and BB increased by 50% and 60%, respectively. However, BD experienced a decrease of 14% in canopy height (Figs. S1, S1A–S1D).

Hurricane Fiona and PUs

The impact of Hurricane Fiona on our restoration project was generally moderate. In total, only four PUs mortality were attributed to Hurricane Fiona; two in BA and one in BB and BC. The total PUs area was reduced by 30.34% after Hurricane Fiona’s onslaught, however, the PUs area reduction was not evenly distributed. For instance, PUs from BA decreased their area by 17.32%, BB by 9.96%, and BD by 135.36%. In contrast, PUs from BC showed an increase in area of 41.27%. Our results also showed a decrease in the daily area expansion of PUs when comparing pre- and post-Fiona estimates i.e., July 2022 and February 2023 monitoring. For block BA, χ2 = 5.124, df = 1, p-value = 0.0236. For block BB, χ2 = 5.967, df = 1, p-value = 0.0146. For block BC, χ2 = 0.773, df = 1, p-value = 0.379, and for block BD, χ2 = 9.8, df = 1, p-value = 0.0017. Likewise, PUs cover showed a slight decrease when comparing pre-and-post Hurricane Fiona monitoring. By contrast, the shoot density after Hurricane Fiona increased compared to before the hurricane.

Discussion

Seagrasses are considered one of the most important coastal habitats in the world. In the Caribbean, the value of seagrass in terms of its ecosystem services is estimated at US $255 billion/year (Shayka et al., 2023). In Puerto Rico alone, seagrass is valued at over US $1.5 billion annually. Additionally, seagrass in Puerto Rico stores the equivalent of US $1 billion worth of carbon, based on California’s (USA) cap and trade program market value for carbon (Costanza et al., 2014; Guerra-Vargas, Gillis & Mancera-Pineda 2020; Pendleton et al., 2012; Fourqurean et al. 2023; Shayka et al., 2023). However, these invaluable ecosystems are currently facing numerous threats that jeopardize their existence. Biotic factors such as overfishing, pollution, coastal development, and abiotic events like storms, heatwaves, and sea-level rise have led to the degradation and loss of seagrass meadows in Puerto Rico and other parts of the world (Johansson, 2002). Thus, given seagrass meadows’ immense ecological and economic value, restoring these habitats and actively ensuring their continued functionality is crucial.

Our restoration strategy clearly shows that planting PUs has successfully restored areas that have experienced physical damage, such as storm impacts and vessel anchoring. Since August 2021 until May 2022, we planted a total of 688 PUs, with an impressive survival rate of 95%. The planted PUs effectively covered an estimated area of nearly 82 m2. These PUs have exhibited a significant expansion rate, with a daily increase of 11.36 cm2/day. Since their initial planting, they have produced approximately 48,000 new shoots, the majority of which belong to the climax seagrass species T. testudinum (Zieman, 1982). The PUs have produced new shoots and facilitated the sexual reproduction of seagrass meadows, as multiple flowers, fruits, and seeds were observed in the PUs, contributing to the enhancement of genetic diversity within the meadow (Fig. S2).

Interestingly, Hurricane Fiona showed a contrasting effect on our restoration project. On a positive note, the storm waves and sediment movement caused by the hurricane resulted in the removal or suffocation of stands of the invasive seagrass species H. stipulacea near the PUs in BD. This rendered a reduction in competition between the native seagrass and the invasive species, allowing the PUs in BD to experience growth levels similar to the other planting blocks that were not affected by the invasive vine. However, the hurricane also had detrimental effects on the PUs. The sediment bedload carried by the storm led to the burial and suffocation of most, if not all, of the recorded PUs mortalities. This was especially impactful for the recently planted PUs that did not have deep anchoring roots to withstand the storm surge. Furthermore, Hurricane Fiona caused a reduction in the extension rate of PUs and in their cover, indicating physical damage from the storm. It is worth mentioning that the loss of tags due to the hurricane may have hindered our ability to fully understand the effects of this disturbance on the PUs’ cover, shoot density, and canopy, although survival rates were still determinable.

On the other hand, we observed that H. stipulacea quickly invaded the PUs from BD and took over the lower portion of the canopy (Fig. S3). This dominance by H. stipulacea may have prevented the native seagrass T. testudinum from slowing the production of new shoots or/and increased their mortality by shading them. However, no PUs mortality was recorded due to competition between H. stipulacea and T. testudinum.

Comparing our results with other peer-reviewed studies or technical reports focused on restoring seagrass beds using T. testudinum PUs is challenging due to their limited availability in sub-tropical Atlantic and Caribbean waters. For instance, in a restoration project conducted by Fonesca, Kenworthy & Thayer (1987) in Florida, USA involving T. testudinum PUs, the survivorship ranged from 0% to 88% over a three-year period. Meanwhile, Uhrin et al. (2009), also working in Florida, reported an 88.9% survival rate for T. testudinum PUs after three years. However, these authors observed that in many cases, surviving PUs exhibited lower shoot densities than when initially planted. On the other hand, Fonesca, Kenworthy & Thayer (1987) estimated the generation of 14 shoots within a 600-day timeframe, which aligns with our own shoot generation estimates. In contrast, Uhrin et al. (2009) reported a modest 11.8 m2 net increase in area and observed no significant expansion in the PU area compared to the initial transplantation stage.

However, our results are consistent with other seagrass restoration studies around the world using different seagrass species. For instance, reported a 95% survival rate after restoring an area of 1,300 m2 in Old Tampa Bay, US, using 400 cm2 plugs of S. filiforme. Bastyan & Cambridge (2008) restored two seagrass meadows in Australia, one at Oyster Harbour and the other at Royal Harbour. They planted 883 units of Posidonia australis with over 71% survival. In New Zealand, Matheson et al. (2017) used mats of 10 cm2 by 15 cm deep and successfully restored an area of 4,200 m2.

Conclusions

Research suggests that restoring meadows using T. testudinum dominated PUs is often time-consuming, typically requiring a minimum of 5 years to observe any noticeable, progress, and expensive (Thorhaug, 1974; Fonesca, Kenworthy & Thayer, 1987; Uhrin et al., 2009). However, our project has demonstrated that meadow rehabilitation with T. testudinum PUs can be feasible in terms of both time and cost. Our successful strategy involved collecting PUs of considerable size, which maximized the netted root and rhizome systems, thereby enhancing the overall survival and growth of the PUs. The presence of a diverse underground environment, including a variety of fauna and flora, also contributed to our positive outcomes.

Nevertheless, there are limitations to using PUs as a restoration strategy. For instance, careful consideration must be given to the number of PUs taken from a single bed, as the removal of PUs can create scars that may take several months to recover. To aid in the recovery process, filling the pits with sandbags, coral rubble, and sand, and periodically removing unwanted organisms can help expedite healing. Similarly, collecting PUs several meters apart does not guarantee genetic diversity. To ensure genetic variation, the planting of seeds is the preferred method. However, it is important to note that seedlings may take multiple years to develop multiple shoots.

Furthermore, before initiating the restoration process, it is crucial to assess the prevailing environmental conditions of the target seagrass bed. Factors such as the frequency of environmental disturbances and the presence of invasive species should be taken into account, as they can induce stress or outcompete the PUs, potentially affecting the overall success of the restoration project.

It is of utmost importance to safeguard seagrass meadows for the sustainable future of both the meadows and the ecosystems surrounding them. Through restoration efforts, we can foster connectivity with other ecosystems, like coral reefs, providing a nurturing environment for numerous commercial fish species that depend on seagrasses during their early stages of life. Protecting seagrass meadows will benefit the shorelines and positively impact the lives of the local community.

Supplemental Information

Supplemental Information 1 The raw measurements for monitoring the Thalassia testudinum and Siryngodium filiforme seagrass conservation project.

Variables were measured in each of the marked PUs over time to monitor the success of the restoration project. Statistical analysis was conducted on this data to compare planting or blocks over time regarding PUs area, cover, and shoot density.

Click here for additional data file.

Supplemental Information 2 PU canopy by Block through time by blocks.

(A) The image showed Block A. (B) showed Block B. (C) Showed Block C. (D) Showed Block D.

Click here for additional data file.

Supplemental Information 3 Different stages of Thalassia testudinum in the PUs.

(A) male flower, (a) A close-up of the male flower. (B) The image shows the female flower of T. testudinum. (C) The image shows the fruit of T. testudinum. (D) A freshly germinated seed of T. testudinum.

Click here for additional data file.

Supplemental Information 4 Plug Unit Propagation (PU) between Halophila stipulacea.

(A. PU was planted between the Halophila stipulacea area in May 2022. (B) Thalassia testudinum surrounded by Syringodium filiforme in April 2023.

Click here for additional data file.

We thank Pedro Gomez, Lemuel Diaz, Ileana Calderon, Mineris Figueroa, Julimar Nevarez-Melendez, Jeremy Velazquez, and Carmen Zayas for their fieldwork assistance in helping establish the Plug Unit Propagation (PUs). Also, we want to thank Marcos Alejandro Quiñones for the map design.

Additional Information and Declarations

Competing Interests

Author Contributions

Field Study Permissions

Data Availability

The authors declare that they have no competing interests.

Claudia Patricia Ruiz-Diaz conceived and designed the experiments, performed the experiments, analyzed the data, prepared figures and/or tables, authored or reviewed drafts of the article, and approved the final draft.

Carlos Toledo-Hernández conceived and designed the experiments, performed the experiments, analyzed the data, authored or reviewed drafts of the article, and approved the final draft.

Juan Luis Sánchez-González performed the experiments, authored or reviewed drafts of the article, and approved the final draft.

Alex E. Mercado-Molina conceived and designed the experiments, analyzed the data, authored or reviewed drafts of the article, and approved the final draft.

The following information was supplied relating to field study approvals (i.e., approving body and any reference numbers):

The Puerto Rico Department of Natural and Environmental Resources (Number DRNA: 2020-IC-020; O-VS-PVS15-SJ-01104-03122019).

The following information was supplied regarding data availability:

The raw data is available in the Supplemental File.

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
