# Peer review of "A successful method to restore seagrass habitats in coastal areas affected by consecutive natural events"

_PeerJ, doi:10.7717/peerj.16700_

## Round 0.1 · original submission · Major Revisions

Please incorporate all comments of reviewers and please submit the revision with a point-to-point rebuttal letter. Please improve the language issues in the paper. The suggested citations are not necessary to incorporate. These can be considered if really needed.

**Language Note:** The Academic Editor has identified that the English language must be improved. PeerJ can provide language editing services - please contact us at copyediting@peerj.com for pricing (be sure to provide your manuscript number and title). Alternatively, you should make your own arrangements to improve the language quality and provide details in your response letter. – PeerJ Staff

Reviewer 1 ·

Basic reporting

The manuscript demonstrates a commendable level of writing proficiency and overall clarity. Nevertheless, there exist select instances where language refinement would enhance both clarity and precision. Furthermore, certain sentences or statements may benefit from rephrasing to optimize the manuscript's overall flow and readability. The figures and tables included in the manuscript are notably well-constructed, appropriately labeled, and effectively bolster the presentation of data. However, it is imperative to ensure that all figures and tables receive appropriate references and discussion within the text. Minor grammatical errors, typographical mistakes, and punctuation issues have been identified at various points throughout the manuscript. We recommend a meticulous proofreading process to rectify these linguistic inaccuracies.

1: In the abstract, briefly mention the implications of the study findings. What do the successful results of this project mean for future seagrass restoration efforts or broader conservation initiatives?

2: Ensure consistency in the use of units of measurement. For instance, you mentioned "900cm2" and later discuss percentage increases. Ensure that these units align correctly throughout the abstract.

3: When mentioning Hurricane Fiona as the primary cause of seagrass mortalities, it might be helpful to briefly elaborate on the impact of the hurricane and why it led to these mortalities.

Introduction
1. The introduction effectively highlights the challenges facing seagrass restoration compared to coral reef restoration. However, consider expanding on the ecological significance of seagrasses in addition to their aesthetic differences with coral reefs. Emphasize their role as critical ecosystems for various marine species and their contributions to carbon storage and coastal protection.

2. When discussing the factors hindering seagrass restoration, it would be beneficial to briefly explain how these factors, such as slow growth rates, affect the restoration process. This would provide more context for readers unfamiliar with seagrass restoration challenges.

3. Instead of using phrases like "from a public view" and "less attractive for funding," consider providing concrete examples or statistics to illustrate the differences in funding levels between coral reef and seagrass restoration projects. This would strengthen your argument.

4. The transition from discussing the challenges of seagrass restoration to its importance is abrupt. Consider providing a smoother transition to explain why despite these challenges, seagrass restoration is crucial.

5. When presenting the ecological and socio-economic benefits of seagrass meadows, it would be helpful to cite specific studies or data to support these claims, adding credibility to your arguments.

6. In the description of your study, briefly mention the goals or objectives of the pilot project. What were you specifically trying to achieve with the restoration efforts in the Canal de Luis Peña Natural Reserve?

7. Provide a concise overview of the impacts of Hurricanes Irma, María, and Fiona on the seagrass meadows in CLPN. Include information on the extent of damage and the areas affected to give readers a clear picture of the situation.

8. When discussing the challenges faced during the post-hurricane recovery, elaborate on how the slow growth of T. testudinum and the expansion of H. stipulacea affected restoration efforts. What were the specific difficulties encountered?

9. Mention the relevance of your study in the context of addressing these challenges and the potential contribution of your pilot project to seagrass restoration efforts in the region.

Material and methods

1. When describing the outplanting process of propagation units (PUs), consider providing more context regarding the size and quantity of PUs used. This could help readers visualize the scale of the restoration effort.

2. Clarify the significance of the modifications made to the PUs harvesting method based on Fonseca et al. (1987). Explain how these modifications improve the chances of survival and growth of PUs, and if possible, provide a concise description of these modifications.

3. In the section discussing the collection of PUs from donor seagrass meadows, provide details about the geographical locations of these donor meadows. Mention whether they were within the same reserve or in different areas, and explain the rationale behind selecting these specific donor sites.

4. When discussing the tagging of PUs, provide information on the specific variables or data that were recorded and monitored using the plastic numbered labels and metallic stakes. This will help readers understand what demographic information was collected.

5. Ensure consistent formatting for citations (e.g., "Fonseca et al., 1987") throughout the Methods section.

Results and Discussion section
1. While emphasizing the economic and ecological value of seagrasses, consider elaborating on how this restoration project aligns with broader conservation goals and strategies. Discuss the project's role in addressing the specific threats mentioned and contributing to the preservation of seagrass meadows.

2. In the discussion of the restoration strategy, elaborate on the potential long-term impacts of the restoration beyond the immediate success observed. How might this restoration affect the surrounding ecosystem, fisheries, and local communities?

3. When discussing the expansion rate of the planted PUs, provide some context or comparison to help readers understand the significance of this rate. For example, you could compare it to the natural growth rate of seagrasses in the absence of restoration efforts.

4. In the section about Hurricane Fiona's effects, consider providing more details on the extent of the hurricane's physical damage and the specific challenges it posed to the restoration project. This could help readers better grasp the complexities of seagrass restoration in the face of natural disturbances.

5. When comparing the results of this restoration project with other studies, mention the similarities and differences in the methods and contexts of these studies. Discuss how these variations might impact the interpretation of the results and their applicability to different restoration scenarios.

Conclusion

1. Consider briefly mentioning potential mitigation strategies or adaptive management approaches that can help address the challenges posed by environmental conditions, disturbances, and invasive species. This would provide readers with practical insights for future restoration efforts.

2. Emphasize the importance of ongoing monitoring and adaptive management in seagrass restoration projects. Discuss how regular assessment and adjustment of restoration strategies can enhance the chances of long-term success and resilience in restored meadows.


3. Conclude with a forward-looking statement about the broader significance of seagrass restoration in the context of marine conservation and the potential benefits for coastal communities, biodiversity, and ecosystem services. This can leave readers with a sense of the broader impact of such initiatives.
4. The section is generally well-organized and provides a clear overview of the introduction, experimental area, design, and methods. However, there are some areas where clarity can be improved and could be benefited from additional detail or references.
 The impact of melatonin on abiotic stress synergies in horticultural plants via redox regulation and epigenetic signaling. Scientia Horticulturae. https://doi.org/10.1016/j.scienta.
 doi: 10.3389/fnut.2023.1167805
https://doi.org/10.3390/agronomy12102584
https://doi.org/10.1002/jsfa.9977
https://doi.org/10.1080/01904167.2022.2056483.
 DOI: https://doi.org/10.2478/fhort-2020-0023 2020-0023

Experimental design

no

Validity of the findings

no

Additional comments

no

Reviewer 2 ·

Basic reporting

1. The paper is poorly written in English, with many grammatical, spelling, and formatting errors. I listed few examples in the abstract and introduction part. The English language should be polished to ensure that an international audience can clearly understand your text.
e.g. In the second sentence of abstract, the authors wrote “three major hurricanes struck the seagrass beds of Culebra, Puerto Rico, causing a 10% decline”. The authors should specify what declined by 10%, such as seagrass cover, biomass, or diversity. The authors should also provide a reference to support this claim.
In the fourth sentence of abstract, “688 Propagation Units (PUs) of the 900cm2 native seagrass”. The authors should use spaces between numbers and units. The authors should also specify what species of native seagrass you used for the PUs.
Line 41, “and, from a public view, harbor less biodiversity”. The comma makes the sentence too long and complex. The authors should use a semicolon instead of a comma to separate two independent clauses that are closely related.
Line 46, “which, when combined, compromises the success of the restoration project, and consequently make the seagrass restoration less attractive for funding”. Use "makes" instead of "make" to be consistent with "compromises".
Line 66, “reduce the seagrass bed community by nearly 10%” the authors should specify what was reduced by 10%, such as seagrass cover, biomass, or diversity.

2. The paper has some unclear or ambiguous terms that need to be defined or explained. For example, "PMEL". The paper should avoid jargon or acronyms that are not familiar to a general audience or provide definitions or explanations for them.

3. The paper does not adhere to the PeerJ policies, as it does not provide a clear and concise abstract, a list of keywords, and a data availability statement.

Experimental design

1. The paper does not provide a clear rationale for choosing the restoration site, the donor sites, and the planting design. It is not clear how the site selection was based on ecological criteria, such as the historical presence of seagrasses, the environmental conditions, and the potential threats. Similarly, it is not clear how the donor sites were chosen to ensure genetic diversity and minimal impact on the source populations. The paper should explain the criteria and methods used for site selection and provide references to support them.

2. The paper does not provide sufficient details on the monitoring methods and data analysis. It is not clear how the covered area, the number of shoots, and the length of leaves were measured and calculated. It is also not clear how the survival rate was estimated and what criteria were used to define mortality. The paper should describe the methods in more detail and provide references to support them. The paper should also report the statistical tests used to analyze the data and provide the p-values and the level of significance for each test.

Validity of the findings

1. The paper does not discuss the possible sources of error or bias in the results, such as the loss of tags, the uneven growth of PUs, or the impact of Hurricane Fiona. These factors could affect the validity and generalizability of the results and should be acknowledged and discussed.

2. The paper should also mention any limitations or assumptions of the methods used and provide suggestions for future research or improvement.

3. The paper does not provide a clear link between the results and the research questions or hypotheses. It is not clear how the results address the main objectives of the paper, such as the impacts of H. stipulacea invasion, the effects of Hurricane Fiona, and the comparison of ecosystem functioning between native and invasive seagrass beds.

Additional comments

Summary of the research and your overall impression:

The paper aims to report the outcomes of a seagrass restoration project in Culebra, Puerto Rico, after three major hurricanes struck the area between 2020 and 2022. The restoration technique used was Plug Propagation Units (PUs) of native seagrasses Thalassia testudinum and Syringodium filiforme. The paper claims that the PUs had high survival and growth rates, and that they improved the seagrass meadow structure and function.
My overall impression is that the paper is interesting and relevant, as seagrass restoration is an important topic for marine conservation and ecosystem services. The paper provides some valuable data on the performance of PUs as a restoration method in a challenging environment. However, I also have some major concerns that need to be addressed before the paper can be accepted for publication.

---

## Round 0.2 · accepted · Accept

The paper is improved after revision and is accepted for publication.

Reviewer 1 ·

Basic reporting

Accepted

Experimental design

no

Validity of the findings

no

Additional comments

no